# Characterization of Diffuse Groundwater Inflows into Stream Water (Part II: Quantifying Groundwater Inflows by Coupling FO-DTS and Vertical Flow Velocities)

**Hugo Le Lay [1,2], Zahra Thomas [1,*], François Rouault [1], Pascal Pichelin [1] and Florentina Moatar [2,3]**

[1]  UMR SAS, Agrocampus Ouest, INRA, 35000 Rennes, France; lelayhugo@gmail.com (H.L.L.); francois.rouault@agrocampus-ouest.fr (F.R.); pascal.pichelin@agrocampus-ouest.fr (P.P.)

[2]  Laboratoire de GéoHydrosystèmes Continentaux (GéHCO), UPRES EA 2100, Université François-Rabelais, UFR des sciences et techniques, Parc de Grandmont, 37200 Tours, France

[3]  Institut National de Recherche en Sciences et Technologies pour l'Environnement et l'Agriculture (Irstea), RiverLy, Centre de Lyon-Villeurbanne, 69625 Villeurbanne, France; florentina.moatar@irstea.fr

**\***  Correspondence: zahra.thomas@agrocampus-ouest.fr or zthomas@agrocampus-ouest.fr; Tel.: +00-33-2-23-48-58-78

**Abstract:** Temperature has been used to characterize groundwater and stream water exchanges for years. One of the many methods used analyzes propagation of the atmosphere-influenced diurnal signal in sediment to infer vertical velocities. However, despite having good accuracy, the method is usually limited by its small spatial coverage. The appearance of fiber optic distributed temperature sensing (FO-DTS) provided new possibilities due to its high spatial and temporal resolution. Methods based on the heat-balance equation, however, cannot quantify diffuse groundwater inflows that do not modify stream temperature. Our research approach consists of coupling groundwater inflow mapping from a previous article (Part I) and deconvolution of thermal profiles in the sediment to obtain vertical velocities along the entire reach. Vertical flows were calculated along a 400 m long reach, and a period of 9 months (October 2016 to June 2017), by coupling a fiber optic cable buried in thalweg sediment and a few thermal lances at the water–sediment interface. When compared to predictions of hyporheic discharge by traditional methods (differential discharge between upstream and downstream of the monitored reach and the mass-balance method), those of our method agreed only for the low-flow period and the end of the high-flow period. Our method underestimated hyporheic discharge during high flow. We hypothesized that the differential discharge and mass-balance methods included lateral inflows that were not detected by the fiber optic cable buried in thalweg sediment. Increasing spatial coverage of the cable as well as automatic and continuous calculation over the reach may improve predictions during the high-flow period. Coupling groundwater inflow mapping and vertical hyporheic flow allows flow to be quantified continuously, which is of great interest for characterizing and modeling fine hyporheic processes over long periods.

**Keywords:** water temperature; groundwater–stream; inflow quantification; vertical flow velocity

## 1. Introduction

Groundwater has great impact on stream ecology since it supports baseflow throughout the year [1–3] and stabilizes stream water quality [4,5], thus providing habitat for many species [6–8]. In their review of hydrologic exchange, Harvey and Goosef [9] outlined challenges in characterizing hydrologic connectivity, exchange flows, and related hydroecological processes. Tools from

hydrological and ecological communities need to be combined to understand groundwater effects on stream flow and predict its future change. Many techniques such as seepage meters, mini-piezometric analysis [10,11], differential gauging [12,13], and chemical tracing [14,15] have been used to quantify groundwater inflow into streams. Each of these techniques has a specific spatial range, accuracy, and limitations [16–20]. Most have the disadvantage of being punctual, such as seepage meters giving information only for a given radius (<1 m). Moving from a fine scale to a coarser scale requires spatially integrative methods such as tracing in specific wells, which sample a large volume and obtain average concentrations, but do not allow for detailed characterization of spatial heterogeneities. Differential gauging allows for measurements at the reach level (a few m to km) but does not allow for detailed description along the longitudinal profile. Also, it requires additional measurements when considering the effects of tributaries. Among these techniques, temperature has been used to trace interactions between groundwater and surface water [21–23]. Since groundwater discharge affects the heat budget of the stream, its temperature can be modeled by solving a 1D transient advection–dispersion equation (Equation (1)) that requires stream temperature ($T_w$), flow velocity ($V$), dispersion coefficient ($D_L$), net heat flux ($\varphi$), specific heat of water ($C_w$), the density of water ($\rho_w$), and the average water column ($d_w$). Variable t is the time and z is the distance along the direction of flow.

$$\frac{\partial T_w}{\partial t} = -V \times \frac{\partial T_w}{\partial t} + D_L \times \frac{\partial^2 T_W}{\partial z^2} + \frac{\varphi}{C_w \times \rho_w \times d_w}. \tag{1}$$

The net heat flux term includes the processes related to stream discharge, atmosphere, groundwater inflow, and streambed conduction. Unidimensional analytical solutions of Equation (1) have been developed and used for decades to infer groundwater inflow from thermal profiles in the sediment [24]. These solutions use the attenuation and phase-shift of the atmosphere-influenced diurnal temperature signal in the sediment to infer vertical flow velocity in porous media, such as the hyporheic zone [25]. Benefiting from the development of affordable, yet precise, thermal sensors, the method has been used in various ways for many years [26,27]. Despite having good accuracy, the method suffers from a limited spatial range; since it is based on vertical thermal profiles in the sediment, it is punctual and usually unable to describe the spatial heterogeneity of inflows [28,29].

The recent use of fiber optic distributed temperature sensing (FO-DTS) has provided new opportunities [30]. Unlike other techniques, FO-DTS provides continuous measurements over space and time at high resolution. Depending on the brand, set-up, and configuration, this technique provides direct measurements with accuracy as high as 0.01 °C and sampling every 0.25 m along cables a few km long [31–33]. Taking advantage of the thermal contrast between groundwater and surface water during summer or winter, FO-DTS has been used mainly to characterize heterogeneous streams and locate focused groundwater inflows [34–36]. Many authors quantified in-stream measurements robustly at multiple times throughout the year [17,37,38]. However, the method is adapted for focused inflows that equal at least 2% of total stream discharge [39], as used in Part I of this study [40]. If groundwater inflows are weak and diffuse, FO-DTS cables must be buried in the sediment to prevent the signal from being displaced by stream flow [41].

To address the challenge of quantifying weak inflows at a high spatial resolution, Mamer and Lowry [42] used FO-DTS to calculate vertical flow velocities of groundwater inflows through the hyporheic zone. They installed two cables at two depths in a controlled flume and were able to estimate vertical flows with relatively good accuracy for uniform inflows. Their system slightly underestimated discharge for focused inflows, however, due to the low spatial resolution of their FO-DTS system (>1 m). Since FO-DTS measurements average the signal over a given distance, misalignment of the cable with the inflow can decrease flow estimates greatly (<25%). It was therefore hypothesized that higher spatial resolution could improve results. However, installing two fiber optic cables at two depths in a natural stream remains challenging, since fast surface flow can disturb the streambed over time [43].

Due to the spatial variability of groundwater inflows, information obtained from punctual (a few cm) or integrative (a few km) methods are difficult to interpret. Multi-scale approaches combining multiple measuring methods may considerably constrain estimates of fluxes between

groundwater and surface water [44–46]. In part I of this two-part study [40], we developed a framework to locate and map groundwater inflow along a reach. Here, we focus on quantifying groundwater inflows, which is essential for investigating resilience of aquatic ecosystems to climate change [47–49]. In this research, vertical flows in the hyporheic zone along the stream were calculated to infer groundwater discharge into the stream. Vertical flow velocities were estimated by replacing the shallowest cable in Mamer and Lowry's study by a few punctual measurements at the stream-sediment interface [42]. Measurements were made at a high spatial resolution in a 400 m long reach in a second-order natural stream. Results from coupling temperature from DTS and from vertical velocities were compared to differential gauging performed over the entire reach and to discharge from heat-balance equations [50], which are more integrative over space. We discuss the potential and relevance of each method. The present article applied the framework presented in the previous article [40], which was dedicated to automatic mapping of such diffuse inflows with sub-hour resolution using FO-DTS. Our research framework consists of two parts to map and quantify diffuse and intermittent groundwater inflows into the stream.

## 2. Materials and Methods

### 2.1. Hydraulics of the Study Site

The geomorphology of the study site was presented in the previous article [40]. The reach actually monitored was 614 m long, however a quantification approach was applied between the two gauging stations because a tributary is located immediately downstream of the second gauging station (Figure 1). Thus, the following part of the study focused on the reach between the upstream and downstream gauging stations; it has no tributary and is 400 m long.

Stream water levels were measured at the gauging stations using OTT Orpheus-mini sensors (accuracy: ±0.5 cm, time-step: 5 min) and then converted into discharge using pre-existing rating curves. These rating curves were refined by punctual discharge measurements ca. every two months using salt dilution tests [50,51]. The upstream and downstream gauging stations in the monitored reach allowed for differential gauging to infer gains and losses of water between stations. To ensure that only exchanges along the hyporheic zone were considered, we extracted baseflow from hydrograms of discharge using the HYSEP program [52]. Subsurface and groundwater flows [53] were considered to be a single component of stream flow. Thus, upstream baseflow in the wetland sub-reach (Figure 1) was called $Qb_{up}$, while downstream baseflow in the meadow sub-reach was called $Qb_{down}$. The differential gauging $Q_{diff} = Qb_{down} - Qb_{up}$ was calculated to estimate the total losses or gains between the stations. $Q_{diff}$ was then used as a reference for comparison with other quantification methods (described later).

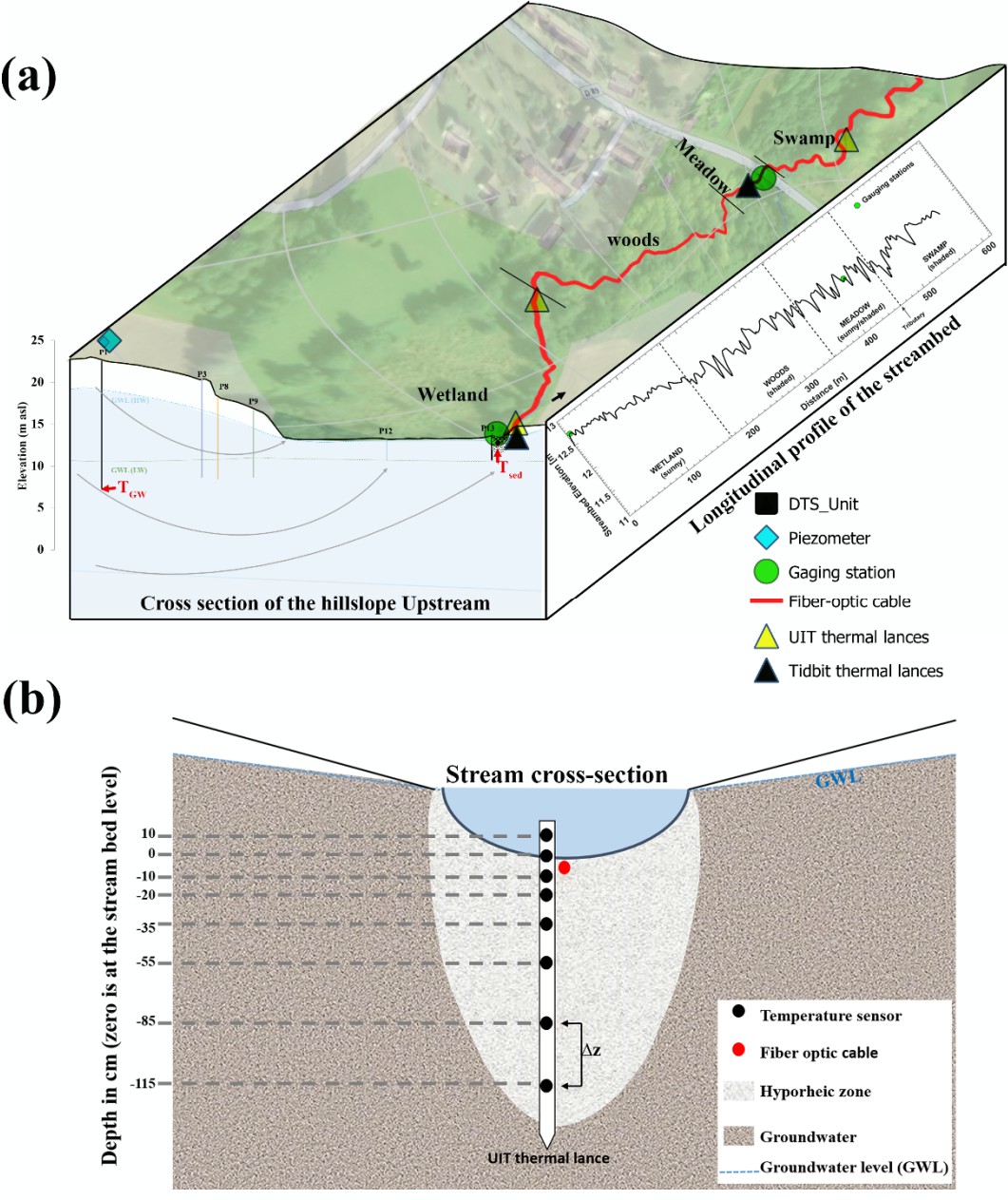

**Figure 1.** 3D view of the study area indicating the hillslope cross section and longitudinal profile. (**a**) Black lines indicate boundaries between sub-reaches with different geomorphologies. The blue diamond indicates the piezometer used to measure groundwater temperature (Tgw) and groundwater level (GWL, dashed blue line). (**b**) Stream cross section showing Umwelt-und Ingenieurtechnik (UIT) lance temperature sensors and depth in the hyporheic zone. Δz is the distance between two sensors used to calculate flow at a groundwater inflow point and a neutral (non-inflow) point, respectively.

### 2.2. Temperature Measurements: DTS Set-up and Thermal Profiles

Thermal profiles in the hyporheic zone [54,55] were obtained using two types of temperature lances. The first were traditional hand-made lances made from a double PVC (Polyvinyl Chloride) tube and fine mesh with TidbiT v2 sensors at four depths: +10, −30, −65, and −100 cm. With an accuracy of ±0.2 °C and a time-step of 5 min, they gave broad estimates of interactions between stream and groundwater in the wetland and immediately before the confluence (Figure 1a). One of the sensors (−30 cm) in the wetland broke down during the campaign, resulting in an incomplete profile. Because of the scarcity of our vertical data and their low spatial resolution and accuracy, three

thermal lances of the second type (Umwelt-und Ingenieurtechnik (UIT) GmbH, Dresden, Germany) were installed. These lances provided thermal profiles with high spatial resolution (+5, 0, −10, −20, −35, −55, −85, and −115 cm) and good accuracy (±0.1 °C) and resolution (0.04 °C) (Figure 1b). These new lances were installed in October 2016, ca. four months after installation of the FO-DTS system. Consequently, the period considered for this study extended from October 2016 to July 2017.

Streambed temperature along the entire reach was measured with an Ultima XT-DTSTM (Silixa Ltd., UK). The cable was buried in the shallow sediment (z ≈ −3 cm) to detect diffuse groundwater inflows despite the stream flow (Figure 1b). The configuration of the set-up, described thoroughly in Part I [40], had an accuracy of ca. 0.05 °C throughout the year with a 40 min time-step and 0.5 m spatial resolution. Groundwater temperature ($T_{gw}$) was measured [56] in the deepest and farthest piezometer on the hillside (Figure 1) using a minidiver submersible pressure transducer (HOBO Pro v2) (accuracy: ±0.2 °C, time-step: 5 min).

### 2.3. Framework for Quantifying Hyporheic Exchanges: Calculating Vertical Flows

All vertical flows were calculated using the VFLUX2 MATLAB toolbox [57]. The program deconvolutes temperature time series between two depths into velocity vectors by solving the 1-D heat transport equation derived from Equation (1) described by [24] for fluid-sediment systems:

$$\frac{\delta T}{\delta t} = K_e \frac{\delta^2 T}{\delta z^2} - q \frac{C_w}{C} \frac{\delta T}{\delta z} \tag{2}$$

where q is the vertical fluid flow in a downward direction (m s$^{-1}$), C is the volumetric heat capacity of the sediment (J m$^{-3}$ °C$^{-1}$), and Cw is the volumetric heat capacity of the water (J m$^{-3}$ °C$^{-1}$). $K_e$ is the effective thermal diffusivity of the sediment, as described by the following equation:

$$K_e = \left(\frac{\lambda_0}{C}\right) + \beta |v_f| \tag{3}$$

where $\lambda_0$ is the thermal conductivity of the sediment (J s$^{-1}$ m$^{-1}$ °C$^{-1}$), β is the thermal dispersivity of the sediment, and $v_f$ is the linear particle velocity (m s$^{-1}$).

To solve Equation (1), VFLUX2 uses, among others, analytical solutions of Hatch et al. [25]. These solutions use the phase-shift and difference in amplitude of the thermal signal between two sensors in the vertical direction to infer fluid velocity:

$$q = \frac{C}{C_w} \left(\frac{2K_e}{\Delta z} \ln A_r + \sqrt{\frac{\alpha + v_T^2}{2}}\right) \tag{4}$$

$$|q| = \frac{C}{C_w} \sqrt{\alpha - 2\left(\frac{4\pi \Delta t K_e}{P\Delta z}\right)^2} \tag{5}$$

where q is the volumetric flow (m$^3$ m$^{-2}$ s$^{-1}$), Δz is the distance between two sensors set in the sediment (m), $A_r$ is the ratio of amplitude between lower and upper sensors (dimensionless), $v_T$ is the velocity of the thermal front (m s$^{-1}$), Δt is the time lag (speed of signal propagation) between temperature signals (s), and P is the period of the temperature signal (s). To obtain a linear velocity v, the following equation is applied:

$$v = \frac{q}{n_e} \tag{6}$$

where $n_e$ is the effective porosity of the sediment (dimensionless). The sign of v determines the direction of the flow: Positive values indicate upward vertical flow (groundwater inflow), while negative values indicate downward flows (point of water loss).

### 2.4. Framework for Quantifying Hyporheic Exchanges: Coupling FO-DTS and Punctual Data

As mentioned, calculating vertical velocity requires at least two temperature measurements separated by a known distance Δz (Equations (3) and (4)). Mamer and Lowry [42] used two fiber optic

cables positioned at two depths to obtain vertical flows with distance [48]. Since this configuration is difficult to control in natural streams, we replaced the shallowest cable—usually placed on the streambed—by punctual measurements of the UIT thermal lances at the water–sediment interface (z = 0 cm) (Figure 1b). This method assumes that (i) punctual temperature at the water–sediment interface is homogeneous in the reach and (ii) the distance between FO-DTS and this interface remains constant (here, $\Delta z$ = 3 cm) over space and time.

Even though our method assumes homogeneous temperatures at the water–sediment interface, we could not ignore that external factors besides hyporheic exchange can modify it locally. For instance, solar radiation can cause local warming in a reach exposed to direct sunlight [58,59], and shading can influence stream temperature [60]. Since the monitored reach had different geomorphological traits (Figure 1a), we subdivided it into four sub-reaches, each of which was attributed a temperature at the water–sediment interface. This subdivision and the attribution of local temperatures was also expected to decrease errors due to the assumption of homogeneity. We also assumed that cable depth remained constant along the entire reach for the entire year of measurements. Obviously, streamflow uncovered it for short periods at some locations, and the streambed moved slightly during the experiment, but these factors were negligible at the reach scale. Uncovered segments of the cable were removed from our dataset when detected, and the cable was found to have remained at generally the same depth when removed in July 2017, after one year of measurements.

Given the thousands of measurements along the FO-DTS, we simplified estimation of vertical flows by selecting only one clear perennial inflow per sub-reach along the FO-DTS cable as a representative inflow point. The vertical velocity calculated for this representative inflow was then considered to represent the entire sub-reach and thus attributed to each groundwater inflow in this sub-reach. Therefore, we made a third assumption: Groundwater flow is homogeneous along a sub-reach. The same approach was applied to locations without inflow (i.e., "neutral"), assumed to be influenced by the atmosphere: One velocity at one representative location was applied to each neutral location along the sub-reach. Selection of these representative locations—inflows and neutral points—was based on groundwater inflow mapping (see Part I [40]). The number and location of inflows and neutral points to which these representative flows were applied were determined from the same map.

The UIT sensor set at the water–sediment interface (z = 0 cm) was used as the upper sensor for each calculation. At this depth, the sinusoidal diurnal signal has large amplitude. The FO-DTS point located above the representative groundwater inflow was used as the lower sensor for calculating flow (Figure 2). At this depth, and because of the groundwater influence, the diurnal signal is clearly dampened ($\Delta A$) and greatly shifted in time ($\Delta t$) compared to that of the upper sensor (Equations (3) and (4)). The representative flow for neutral points was calculated directly using the UIT sensors at z = −10 cm. Its diurnal signal is not influenced by upward groundwater flow, so it is less dampened and phase-shifted. UIT lances were used for neutral points because the groundwater inflow mapping by FO-DTS (Part I [40]) could not completely distinguish true neutral points in the sediment from uncovered cable. In contrast, UIT lances were placed in zones without clear inflow, and the depths of their sensors were known with certainty throughout the year. Since no UIT lances were placed in the meadow (Figure 1), calculations in this sub-reach were based on the second UIT lance placed in the woods. UIT lances were preferred over the older TidbiT lances because of their proximity to the water–sediment interface (first 10 cm) and their greater accuracy (±0.1 °C vs. ±0.2 °C, respectively).

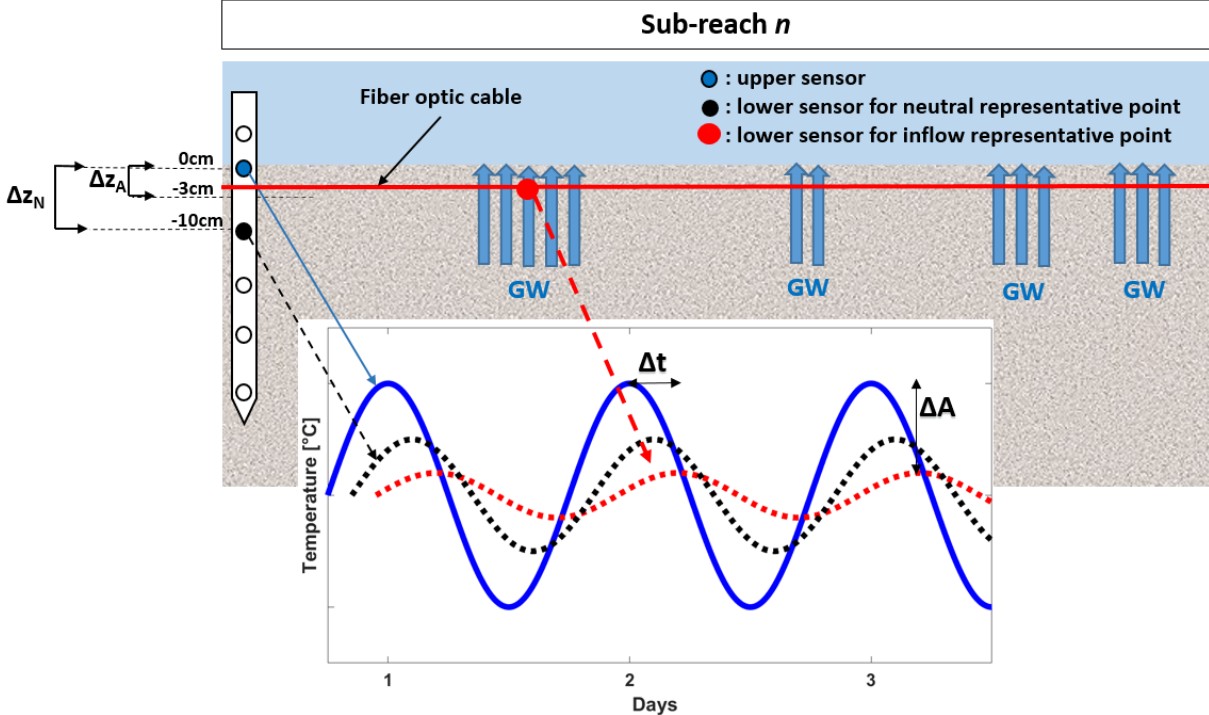

**Figure 2.** Conceptual representation of calculating vertical flow along a sub-reach. The upper sensor used for all velocity calculations is the thermal sensor at the water–sediment interface ($z = 0$ cm). A clear and perennial groundwater (GW) inflow is selected as a representative point and the fiber optic distributed temperature sensing (FO-DTS) point set above it is used as the lower sensor for calculating inflow velocity ($z = -3$ cm). A clear and perennial atmosphere-influenced stream water location was selected in each sub-reach as representative of "neutral" sections. In this study, all thermal lances were set in neutral locations without inflow: their stability in the sediment made them more accurate than the FO-DTS system in calculating the representative neutral flows ($z = -10$ cm).

Calculating vertical velocities based on analytical solution of 1D heat equation assumes that the fluid flow is vertical and one-dimensional. The temperature signal is assumed to be sinusoidal and there is no thermal gradient in sediment between the two sensors in the vertical direction [61]. As discussed in Irvine et al. [61], there is also a limitation regarding stream-bed heterogeneities which affect thermal properties of the sediments. Vandersteen et al. [62] developed a new method to calculate vertical flow called LPML (Local Polynomial Maximum Likelihood). This method was compared to VFLUX which was developed by Gordon et al. [57]. There were also major limitations with analytical solutions to the 1-D heat transport equation since streambed heterogeneity or non-vertical flow components are not considered. In our study, we parameterized input parameters of VFLUX2 (Table 1) based on sediment properties. Thus, a sampling campaign was performed in September 2017 to assess thermal properties of the sediment at multiple points along the stream. Each sub-reach was sampled due to geomorphological differences among sub-reaches. Sediments in the wetland and woods were similar to those in the confluence: dominated by silts and organic matter, with a sand fraction. In contrast, sediment in the meadow was exclusively sandy and relatively shallow. Dry bulk densities of these sediments were measured using the kerosene-displacement method [63]. Based on these densities, the thermal parameters required for calculations were estimated according to Lapham [64]. The standard deviation (SD) of each parameter was determined based on the uncertainty in bulk density measurements Lapham [64]. These SDs were used in 500 Monte Carlo iterations to estimate uncertainty in the vertical velocities calculated.

**Table 1.** Means (and standard deviations, if ≠ 0) of input parameters of VFLUX2 [57] used in this study for each sub-reach.

| Parameter | Symbol (Unit) | Sub-reach | | | |
|---|---|---|---|---|---|
| | | Wetland | Woods | Meadow | Swamp |
| Fundamental period to filter | P (day) | 1 | 1 | 1 | 1 |
| Sediment porosity | $\eta_e$ (-) | 0.43 (0.01) | 0.40 (0.01) | 0.26 (0.07) | 0.40 (0.01) |
| Sediment thermal dispersivity | b (m) | 0.001 | 0.001 | 0.001 | 0.001 |
| Sediment thermal conductivity | $\lambda_0$ (cal s$^{-1}$ cm$^{-1}$ °C$^{-1}$) | 0.0045 (0.0001) | 0.0048 (0.0001) | 0.0069 (0.0001) | 0.0048 (0.0001) |
| Sediment volumetric heat capacity | C (cal m$^{-3}$ °C$^{-1}$) | 0.643 (0.001) | 0.630 (0.001) | 0.560 (0.002) | 0.630 (0.001) |
| Water volumetric heat capacity | $C_w$ (cal m$^{-3}$ °C$^{-1}$) | 1.00 | 1.00 | 1.00 | 1.00 |

*2.5. Framework for Quantifying Hyporheic Exchanges: Comparing Vertical Velocities to Volumetric Discharge*

In a punctual approach, Rosenberry et al. [35] compared their calculated vertical flows to seepage meters and found a strong correlation ($R^2$ = 0.96). Our method aimed to estimate groundwater inflows in a more continuous approach over an entire hydrological year. Since seepage meters were not available and not recommended for such long measurements, we compared our results not only to the differential gauging $Q_{diff}$ but also to the heat-balance method of [37], adapted to the site. Since the FO-DTS cable was buried in the sediment to detect weak inflows (Figure 1b), stream water and groundwater did not mix much at the cable. Moreover, groundwater inflows at the site were expected to be too weak and diffuse to modify the stream temperature. Therefore, it was not possible to apply the heat-balance method to each point. Instead, we used it to estimate groundwater discharge between the 400 m separating the two gauging stations ($Q_{balance}$), in this form:

$$Q_{balance} = \frac{Qb_{down} \times T_{down} - Qb_{up} \times T_{up}}{T_{gw}} \tag{7}$$

where $T_{up}$ and $T_{down}$ are the temperatures at the TidbiT thermal lances at the upstream and downstream gauging stations, respectively.

Since the 400 m over which $Q_{diff}$ and $Q_{balance}$ were estimated covered the first three sub-reaches upstream of the tributary, only vertical flows between the stations were considered for comparison. To compare volumetric discharge measurements along a reach ($Q_{diff}$ and $Q_{balance}$) to linear discrete vertical velocities, we tested two approaches.

First, we converted the vertical flows (every 0.25 m) predicted by VFLUX2 into a volumetric discharge ($Q_{coupling}$) by considering a specific surface (S) for each punctual velocity (v), whether directed upward (groundwater inflow), downward (water loss) or null (neutral):

$$q_{coupling(t,i)} = v_{(t,i)} \times (l \times w) \tag{8}$$

where t is the time step, i is a point of the cable, l the length of the area and w the width of the area.

The length was set to 0.25 m (sampling distance of the DTS), but two widths were tested: 2 m (mean width of the streambed) and 0.5 m (within the 0.30–0.70 m range observed during low flow and equal to the spatial resolution of the DTS). Once $Q_{coupling}$ was calculated for each point, net discharge ($Q_{coupling}$) (gains − losses) along the reach due to interactions between the stream and the aquifer was calculated as follows:

$$Q_{coupling(t)} = \sum_{i}^{end} q_{coupling(t,i)}. \tag{9}$$

In the second approach, we converted the volumetric discharge into velocity ($V_{diff}$) using the following equation:

$$V_{diff} = \frac{Q_{diff}}{w \times L} \tag{10}$$

where L is the distance between the two gauging stations (400 m).

As for the previous approach, two widths were tested: 2 m and 0.50 m.

## 3. Results and Discussion

### 3.1. Groundwater Inflow Mapping

Normalized thermal anomalies mapped using the framework presented in Le Lay et al. [40] showed spatio-temporal variability over the study period (Figure 3). Neutral locations were assumed to be influenced only by the atmosphere (Figure 3b, in blue), unlike groundwater inflow locations (Figure 3b, in yellow). Some anomalies interpreted as inflows could have been due to hyporheic circulation or the cable becoming more deeply buried. Moreover, some inflows did not behave as expected given their location. For instance, strong and stable groundwater inflows were located upstream, in the wetland sub-reach, even though it was perched. The processes involved were suspected to be either groundwater inflows from the nearby hillslope or water drained from agricultural fields. Despite some uncertainty about certain inflows, we assumed here for convenience that all thermal anomalies (yellow zones in Figure 3) were indeed groundwater inflows. All representative groundwater inflows, chosen for their constancy over time, were located in pools.

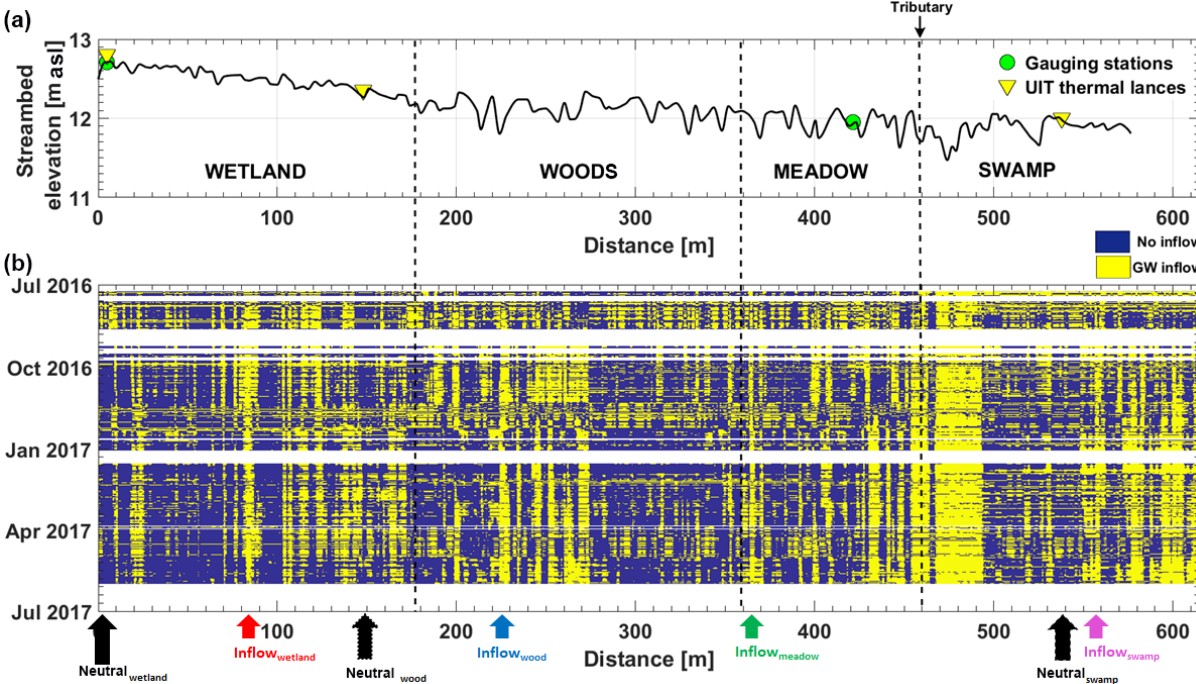

**Figure 3.** (**a**) Longitudinal profile of the streambed in the four sub-reaches of the monitored reach. UIT = Umwelt-und Ingenieurtechnik, m asl = m above sea level. (**b**) Spatio-temporal evolution of thermal anomalies [40]. Arrows indicate representative points of groundwater (GW) inflow and no inflow (Neutral). White strips show periods when the distributed temperature sensing system malfunctioned (horizontal) or points where the cable was often non-submerged (vertical).

### 3.2. Evolution of Thermal Profiles in the Hyporheic Zone

Overall, the temperature of the entire hyporheic zone varied over time and followed the seasonal cycle, with lower temperatures in winter (January 2017) and higher temperatures in summer (June 2017) (Figure 4). Nevertheless, temperature was vertically stratified throughout the study period, and that in the stream (z = +5 cm) varied much more than that at −115 cm (0–21 °C and 7.25–13.75 °C,

respectively). The annual thermal cycle was thus clearly defined along the lances. During autumn and winter, shallower sensors recorded colder temperatures than deeper ones (up to 9 °C colder in January 2017) but with a consistent thermal stratification: all depths had distinctly different temperatures. Thermal stratification was also observed in late spring (May–June 2017), with higher temperatures at shallower depths (up to 8 °C warmer in June 2017). However, from February to early May 2017—the period defined as the peak of high flow in Part I [40]—temperature of the hyporheic zone was relatively homogeneous. Most thermal contrast (0.7–6.0 °C, depending on the day) was observed in the stream water (z = +5 cm) and at the water–sediment interface (z = 0 cm), while the remaining contrasts were small (0–3 °C). Moreover, short periods with little thermal contrast between deep and shallow depths were also observed in mid-November 2016, December 2016, mid-January 2017 and mid-May 2017. Similarly, some periods of even smaller contrast (as small as 0.7 °C) between February 2017 and May 2017 could be attributed to the same processes. This behavior could have been due to a surge of deeper groundwater in the sediment that decreased propagation of the diurnal signal in the sediment. Only thorough study of the phase-shift and difference in amplitude between two vertical locations (Equations (3) and (4)) could provide the amplitude and direction of flow over time in the hyporheic zone. We thus calculated vertical flow velocity from the data at −10 and 0 cm to determine whether this potential surge of groundwater truly discharged into the stream.

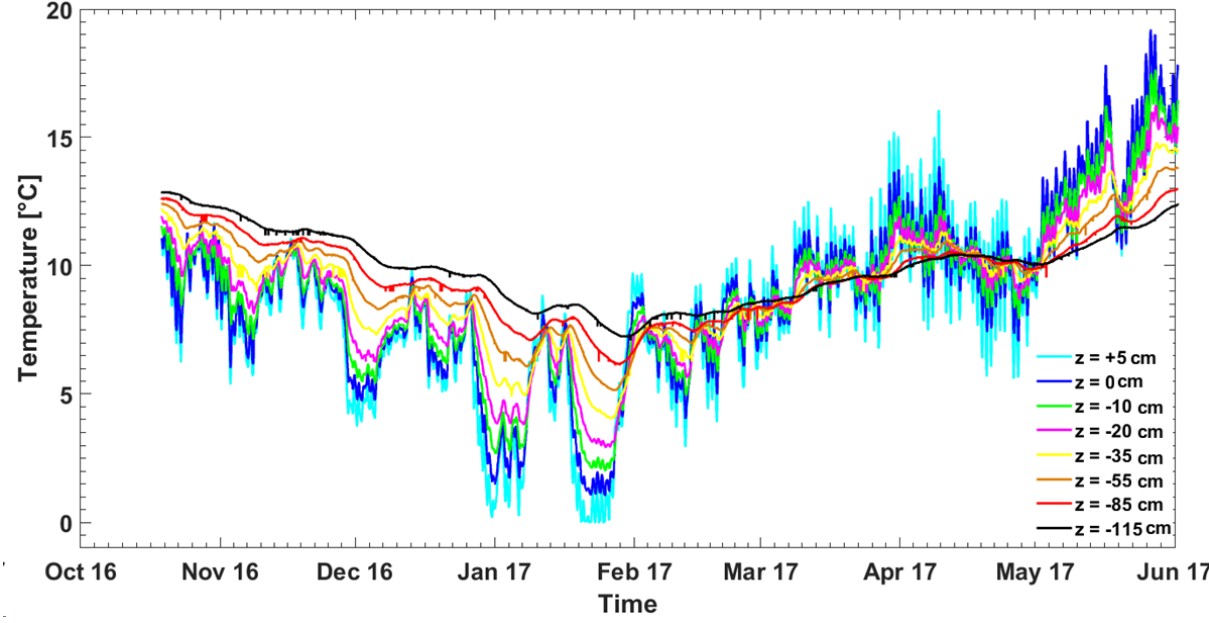

**Figure 4.** Example of temperature profile dynamics along the hyporheic zone from the thermal lance located in the wetland sub-reach.

Two-dimensional thermal profiles along the hyporheic zone at neutral locations revealed great similarity among the three points monitored (Figure 5). Colder surface temperatures propagated into the sediment from October 2016 to February 2017 along all three lances. In contrast, warmer surface temperatures propagated downward in late spring (May–June 2017). In between (February–May 2017), sediment temperatures remained relatively homogenous. Monthly mean profiles also showed quasi-identical thermal envelopes [65], with few differences above the streambed (z = +5 cm) during this period (Figure 5). The annual amplitude of temperature in the wetland, woods and swamp sub-reaches was 12.5 °C, 12.7 °C and 12.4 °C, respectively, at the water–sediment interface (z = 0 cm) and ca. 6.4 °C, 6.5 °C, and 6.2 °C, respectively, at the deepest point monitored in the hyporheic zone (z = −115 cm). Only the swamp (i.e., the most downstream sub-reach) had a slightly thinner thermal envelope (−55 to −115 cm), which indicates a stronger influence of groundwater in this part of the reach. Because the UIT lances were installed before mapping thermal anomalies, they were not placed in locations of clear groundwater inflow. This is consistent with field observations and the inflow mapping, which indicated that the swamp sub-reach was influenced greatly by groundwater.

Overall, propagation of surface temperature through the hyporheic zone was clear, especially in winter from late December to February and in May–June. Temperature propagated faster above 55 cm than below it; however, in mid-December and mid-January, temperature homogenized quickly at 11–12 °C, close to that of groundwater. This thermal signature may suggest sudden groundwater inflow but could also represent downward flows with relatively warm temperature for the season. Whether these episodes were groundwater surges that discharged into the stream or warm downward flows was impossible to determine based solely on these qualitative data. We thus used the persistent difference in temperature between the water–sediment interface (z = 0 cm) and the first depth (z = −10 cm) to infer the vertical flow over time and its direction through the sediment.

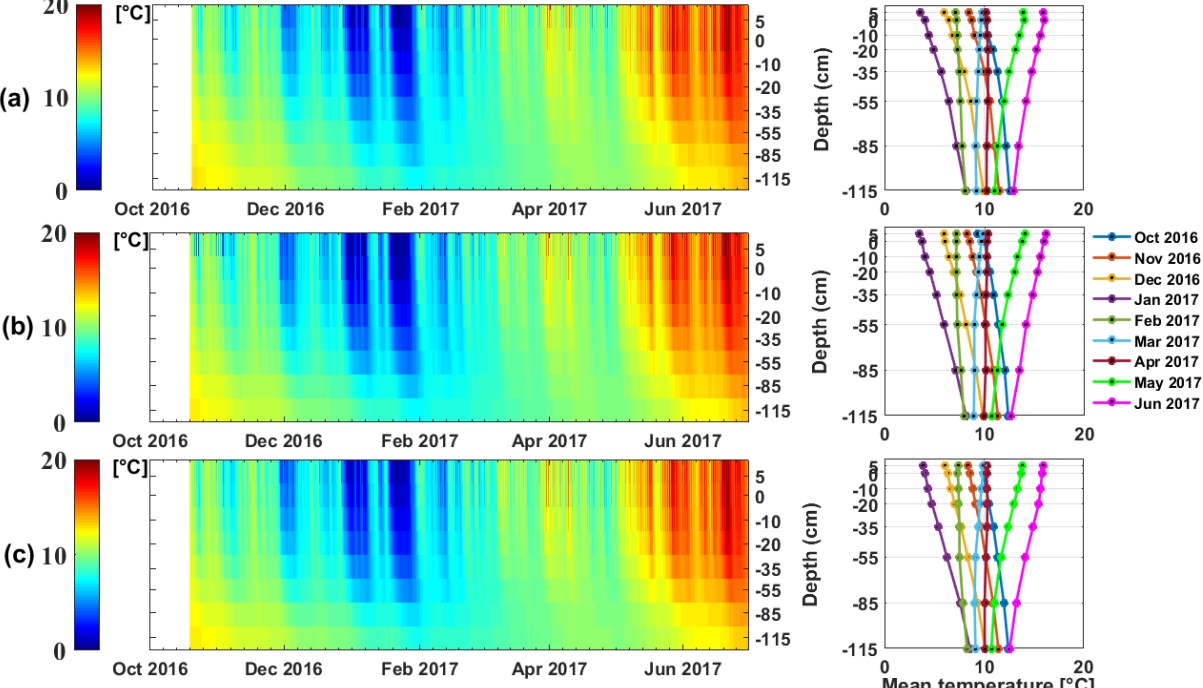

**Figure 5.** Thermal profiles along the hyporheic zone at locations with no groundwater inflows detected (**left**) at a time-step of 5 min or (**right**) as monthly means in the (**a**) wetland sub-reach (upstream), (**b**) interface between wetland and woods sub-reaches, and (**c**) swamp sub-reach (downstream).

## 3.3. Vertical Flow Velocities at Locations without Groundwater Inflows

At neutral locations (i.e., without inflow and assumed to be influenced only by the atmosphere), vertical flow velocities in the wetland, near the woods and in the swamp sub-reaches were generally low (0 to $-1 \times 10^{-5}$ m s$^{-1}$), directed downward (negative values), similar over space and constant over time (Figure 6). Nevertheless, larger downward velocities were observed in the swamp in mid-November 2016, December 2016 and late January 2017 (>ca. $-2 \times 10^{-5}$ m s$^{-1}$). Ultimately, no clear changes in direction (upward flows) or intensity (high velocities) could be related to the short periods when the temperature along the lance varied little. In fact, some of the downward velocity peaks in autumn 2016 occurred at the same time as these suspected groundwater surges. Part I of this study [40] observed flood events occurring at the same time as a weak negative hydraulic gradient at the site from November 2016 to January 2017. It is likely that these sudden floods temporarily inversed the hydraulic gradient and triggered downward flows (water loss).

Ultimately, the groundwater surges observed in autumn and during high flow (Figure 3) did not become groundwater discharges into the stream. With flows generally directed downward at a mean of $-0.4 \times 10^{-5}$ m s$^{-1}$, the neutral locations thus remained neutral most of the studied period. We therefore assumed that this was the case for all points considered neutral. These velocities were thus applied to all neutral points in each sub-reach to calculate global discharge ($Q_{coupling}$).

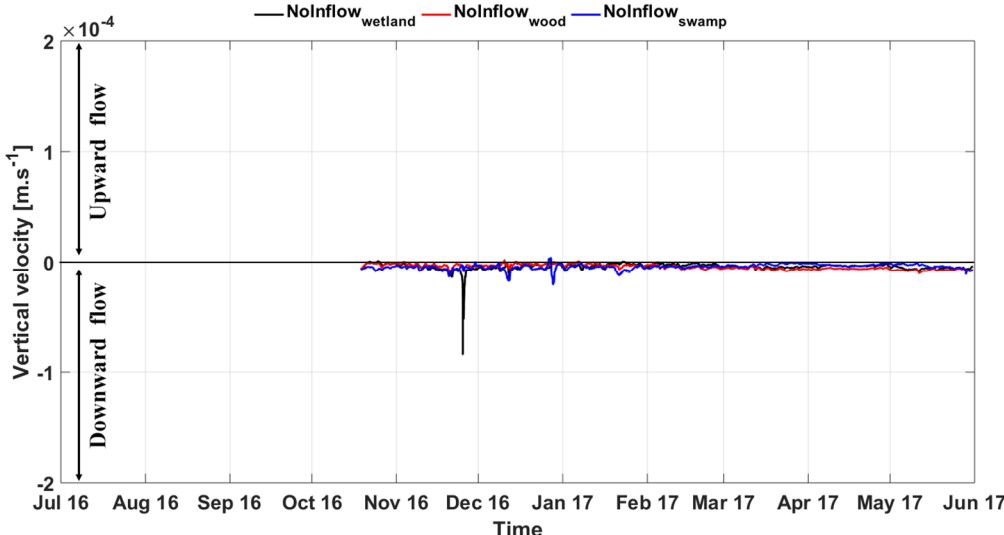

**Figure 6.** Vertical flow velocities along the hyporheic zone from locations without groundwater inflows (Neutral) estimated by VFLUX2 from $z2 = -10$ cm to $z1 = 0$ cm (i.e., negative values indicate downward flows).

### 3.4. Vertical Flow Velocities in Groundwater–influenced Locations

Unlike velocities calculated at neutral points, most of those at groundwater–inflow points were directed upward (positive values), confirming groundwater inflows (Figure 7). Moreover, all were higher than the downward velocities at neutral points by one order of magnitude (up to $1.3 \times 10^{-4}$ m $s^{-1}$ in October and $1.8 \times 10^{-4}$ m $s^{-1}$ in April). Vertical velocities in all four sub-reaches showed similar patterns, ranging from $0.5–1.3 \times 10^{-4}$ m $s^{-1}$ at the end of the low-flow period (October 2016), decreasing from November 2016 to January 2017 (including even downward flows as high as $-0.8 \times 10^{-4}$ m $s^{-1}$) and then increasing again from February–June 2017 ($0.3–1.8 \times 10^{-4}$ m $s^{-1}$). At a shorter time scale, velocities in all sub-reaches increased and decreased simultaneously, differing only in the amplitude of these changes depending on location and time. The swamp always had higher velocities than the other three sub-reaches and never had downward flows, even during the overall decrease from November 2016 to January 2017. Vertical velocity in the meadow sub-reach varied the most over time, while that in the wetland and woods sub-reaches varied less. Velocity in the woods sub-reach decreased from 1.0 to $0.5 \times 10^{-4}$ m $s^{-1}$ in November 2016 and, except for common downward flow episodes in November and December 2016, remained approximately the same until June. Overall, vertical velocity in the wetland sub-reach was lower than those in the other sub-reaches from November-February.

This apparent difference between lower upstream (wetland and woods sub-reaches) vertical velocities and higher downstream (swamp sub-reach) vertical velocities was probably related to stream geomorphology. The upstream streambed's perched elevation probably constantly decreased its hydraulic gradient, explaining its lower velocities. In contrast, field observations had already determined that the swamp experienced regular groundwater surges: Its banks were often flooded or at least wet. Moreover, the swamp had a lower streambed with more pools than the rest of the reach. It thus seems likely that the swamp's higher vertical velocities were due to inflows caused by a higher hydraulic gradient. Velocities in the meadow sub-reach reflected the evolution of thermal anomalies (Figure 3b), being low from November 2016 to February 2017 but increasing greatly during the high-flow period, when the number of locations with inflows also increased. The meadow's greater variability in velocities was most likely due to its sandier sediment and the sensitivity of the VFLUX2 program to its higher thermal conductivity. Effects of sediment characteristics on hyporheic flow have been addressed by many authors [66–68]. The general decrease in vertical flows from November 2016 and February 2017 echoed the few similar episodes detected at neutral locations

(Figure 5). Therefore, this decrease was probably caused by the same punctual gradient inversions following flood events in December.

Ultimately, the flows calculated across the entire reach clearly indicated positive groundwater discharges, showing the relevance of the inflow mapping method developed in Part I [40]. Moreover, despite having different amplitudes, these groundwater velocities were quite synchronized, supporting the assumption of homogeneous groundwater temperature. These velocities were thus applied to each point of their respective sub-reach that was considered to be an inflow in order to calculate global discharge ($Q_{coupling}$), which was then compared to the other quantification methods.

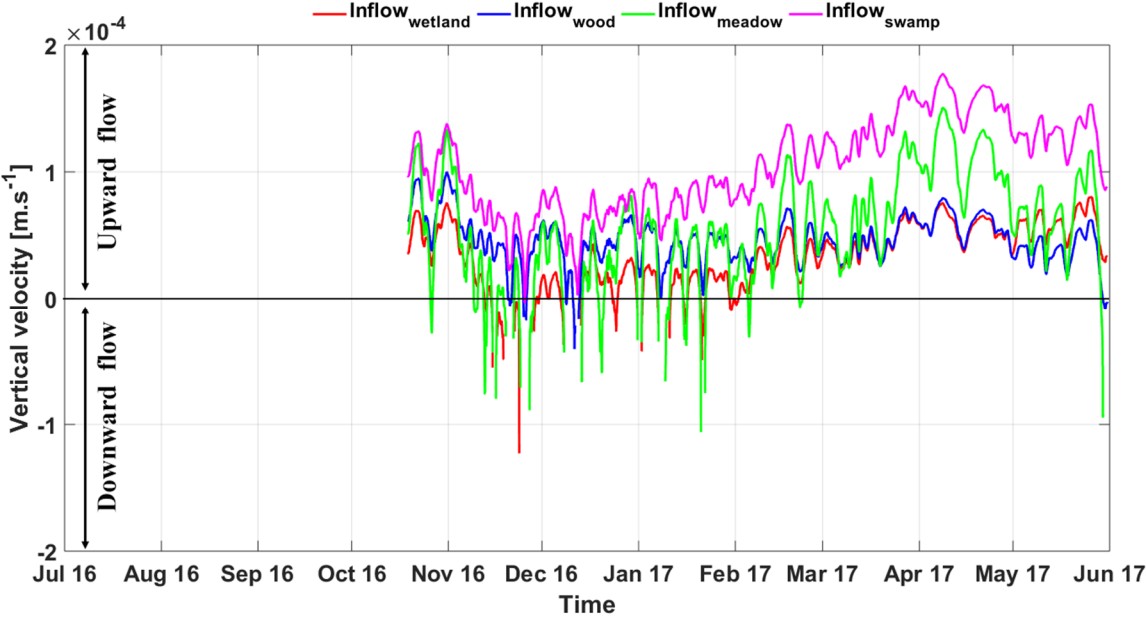

**Figure 7.** Vertical flow velocities calculated from temperatures at z = 0 cm and fiber optic distributed temperature sensing (z = −3 cm) for the wetland, woods, meadow, and swamp sub-reaches. Error bars are based on 500 Monte Carlo iterations with variable thermal parameters of the sediments.

### 3.5. Comparison of Groundwater Discharge Estimation Methods

Discharge dynamics upstream and downstream clearly revealed the low- and high-flow periods previously identified. During the low-flow period (July to late October 2016), stream discharge was generally low upstream and downstream (mean = 22.3 and 24.6 L s⁻¹, respectively) with no large flood events (Figure 8). In contrast, discharge during the high-flow period from November 2016 to June 2017 varied more. The first large flood occurred in November 2016, which may explain the mostly downward flows calculated by VFLUX2 (Figures 6 and 7). During flood events, baseflow increased steadily upstream and downstream, reaching 49 and 109 L s⁻¹, respectively, in March 2017 (Figure 8). Discharge peaks in December 2016 and May 2017 were not considered as groundwater discharge since they were probably still a result of residual runoff.

Upstream and downstream baseflows had similar dynamics, differing only in their amplitudes. Baseflow was clearly higher downstream than upstream during the high-flow period, which supported the idea that groundwater fed the reach during this period. Baseflows during the low-flow period were similar upstream and downstream and lay within the uncertainty in measurements, masking potential water gains or losses.

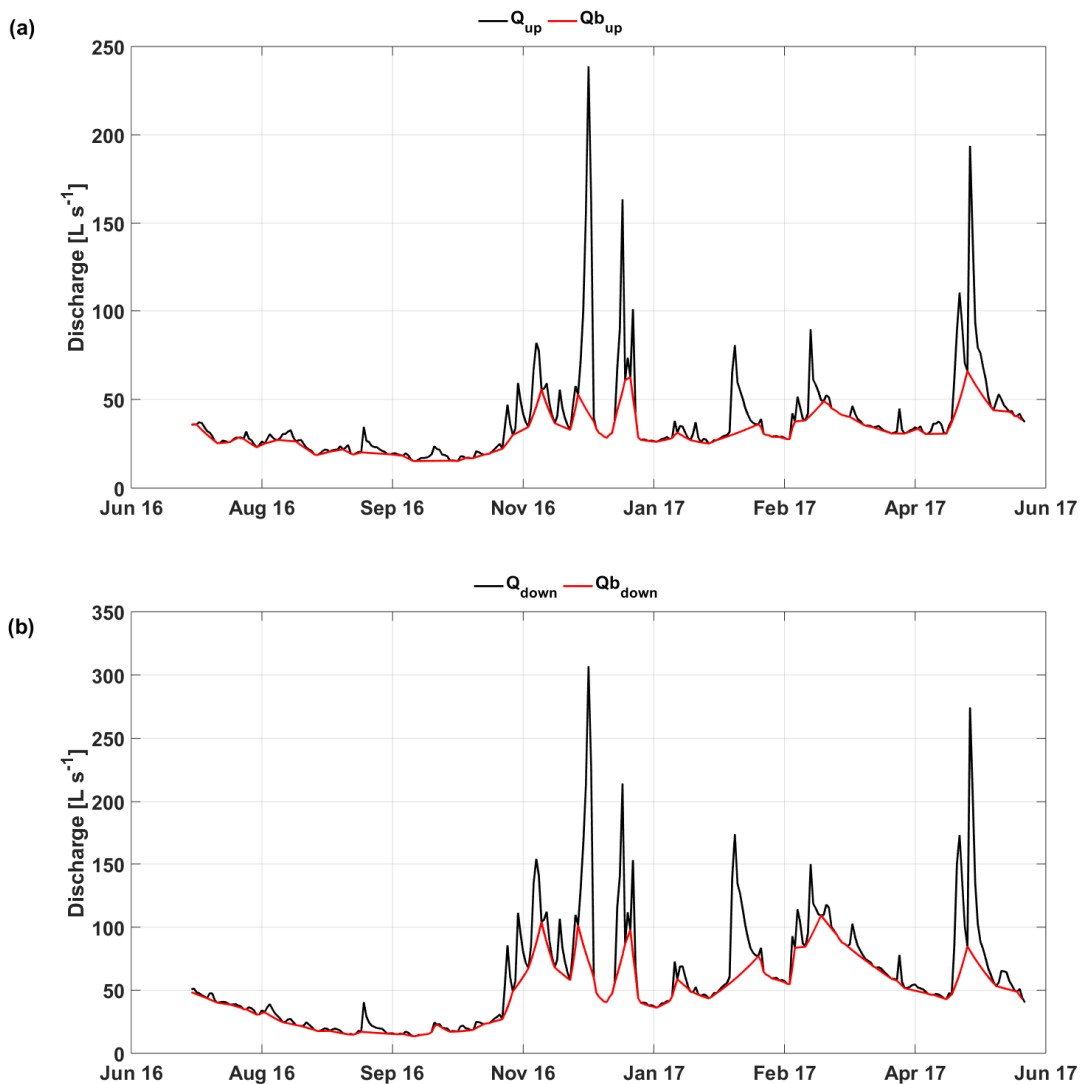

**Figure 8.** Stream discharge (Q) (including runoff) and baseflow (Qb) over time at the gauging station (**a**) upstream (wetland sub-reach) and (**b**) downstream (meadow sub-reach).

Figure 9 shows a comparison of the three methods for quantifying hyporheic discharge i.e., (i) $Q_{balance}$ from conservation of mass and energy, (ii) $Q_{diff}$ from differential gauging, and (iii) $Q_{coupling}$ from coupling FO-DTS and vertical temperature profiles in hyporheic zone. $Q_{diff}$ had weak, almost negative values from August–October 2016 (+8 to −6 L s$^{-1}$) (Figure 9), consistent with measurements during low flow, then increased until March 2017 (up to +62 L s$^{-1}$). Its net decrease after March was attributed to the beginning of root uptake by vegetation and evaporation in the riparian area [69]. $Q_{balance}$ was similar to $Q_{diff}$ except for being higher than it in October 2016 and then lower than it during the high-flow period, especially in January and February 2017 (2–22 L s$^{-1}$ lower). The larger differences were likely due to a lag time between hydraulics and stream temperature: During storms or heavy precipitation, stream water level increases quickly, but stream temperature does so more slowly because of its thermal inertia. This highlights limits of the $Q_{balance}$ approach over large distances: By assessing conservation of mass and energy, the longer the distance between measurements (here, nearly 400 m), the more mass and energy exchanges with the environment become likely, rendering the method less accurate. However, periods with low precipitation and moderate solar radiation (autumn and spring) yielded good results due to the short distance, high flow and shading at the site, which decreased energy inputs.

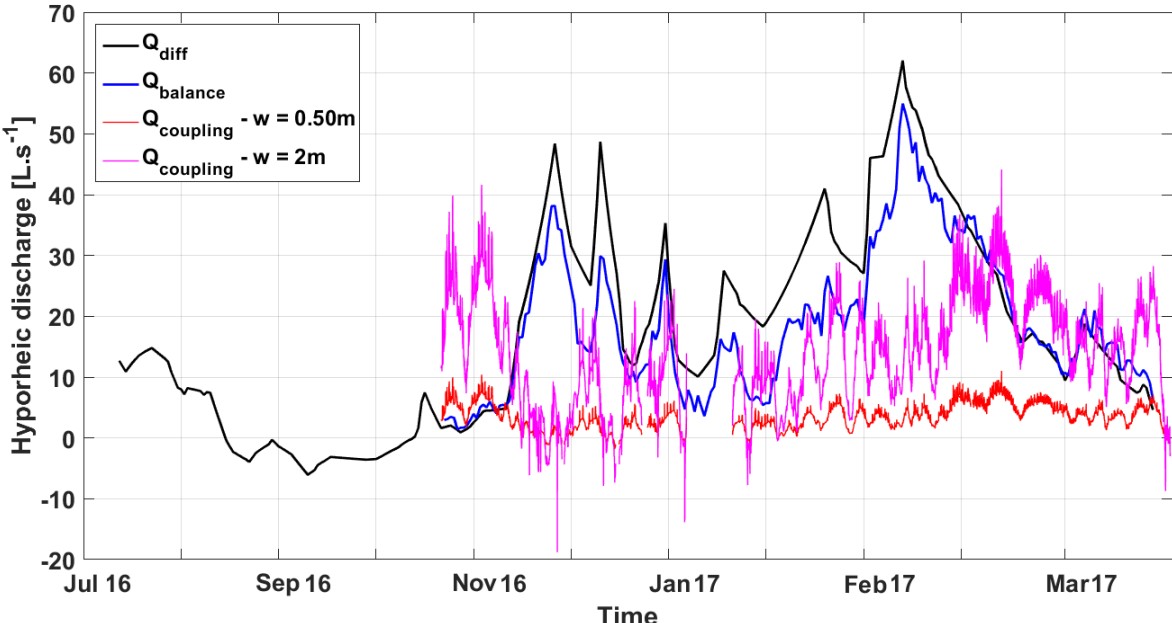

**Figure 9.** Comparison of three methods for quantifying hyporheic discharge along the 400 m long reach between gauging stations: differential gauging ($Q_{diff}$), heat-balance method ($Q_{balance}$), and the method coupling distributed temperature sensing mapping and vertical flow velocities ($Q_{coupling}$).

The hyporheic discharge estimated with our method ($Q_{coupling}$) (gains from groundwater inflows—potential losses from neutral points) depended on the width used to calculate it. For a width of 0.50 m, $Q_{coupling}$ was similar to $Q_{diff}$ during low flow but much lower than $Q_{diff}$ and $Q_{balance}$ during the high-flow period. For instance, during peak discharge in March 2017, $Q_{coupling}$ was nearly 55 L s$^{-1}$ (−90%) lower than $Q_{diff}$. This underestimation was lower in December 2016 and May 2017 but remained high (−65% to −85%). For a width of 2 m, $Q_{coupling}$ overestimated $Q_{diff}$ by 15 L s$^{-1}$ (+700%) during low flow and generally underestimated it during high flow; however, underestimation was smaller, and even close to zero, in May 2017. Hyporheic discharge at the beginning of high flow (November–December 2016) was low or even negative (water loss) for both widths. This pattern was the expression of the downward flows observed at the site (Figures 6 and 7) but was in disagreement with $Q_{diff}$. In this context, the three methods showed consistent estimates of hyporheic discharge during low flow for the narrower width (0.5 m) and the end of the high-flow period. Hyporheic discharge occurring over the entire streambed width (2 m) was also underestimated during the high-flow period. Also, the order of the magnitude of discharge estimated by coupling DTS and the thermal gradient in the sediment is similar to those of groundwater inflows calculated using radon at the tributary outlet in the same study site [70]. In this study, groundwater inflow of ca. 0–5 L s$^{-1}$ was estimated along the Vilqué reach.

Since the hydrological year was relatively dry, the stream did not overflow its banks; consequently, its width never exceeded its minor bed. All three methods considered water gains and losses along the reach. $Q_{diff}$ and $Q_{balance}$ spatially integrated discharge information between the stations, while $Q_{coupling}$ was based on velocities from groundwater inflows and neutral points.

Since the number of groundwater inflows and neutral points varied over time, $V_{coupling}$ was calculated as the mean of groundwater velocities in the wetland, woods, and meadow and velocities above neutral points in the wetland and woods. Comparisons of flow velocities through the streambed ($V_{diff}$ and $V_{coupling}$) yielded results similar to those of hyporheic discharge. During high flow, $V_{coupling}$ was similar to $V_{diff}$ when a width of 2 m (i.e., the streambed) was assumed (Figure 10a). Some differences were observed in late November 2016 and March 2017 but the two velocities were particularly similar at the end of the high-flow period (late April and May). In contrast, during low flow, $V_{coupling}$ was closer to $V_{diff}$ when a width of 0.5 m was assumed. Cumulative probability analysis revealed that the velocities in the hyporheic zone predicted by our method generally had the same

mean and dynamics as those of V<sub>diff</sub> when assuming a width of 2 m (Figure 10b). From these results, the few vertical velocities calculated with FO-DTS seemed representative of overall groundwater flow at the site. Based on estimated velocities, groundwater inflow rates seemed relatively uniform across the site.

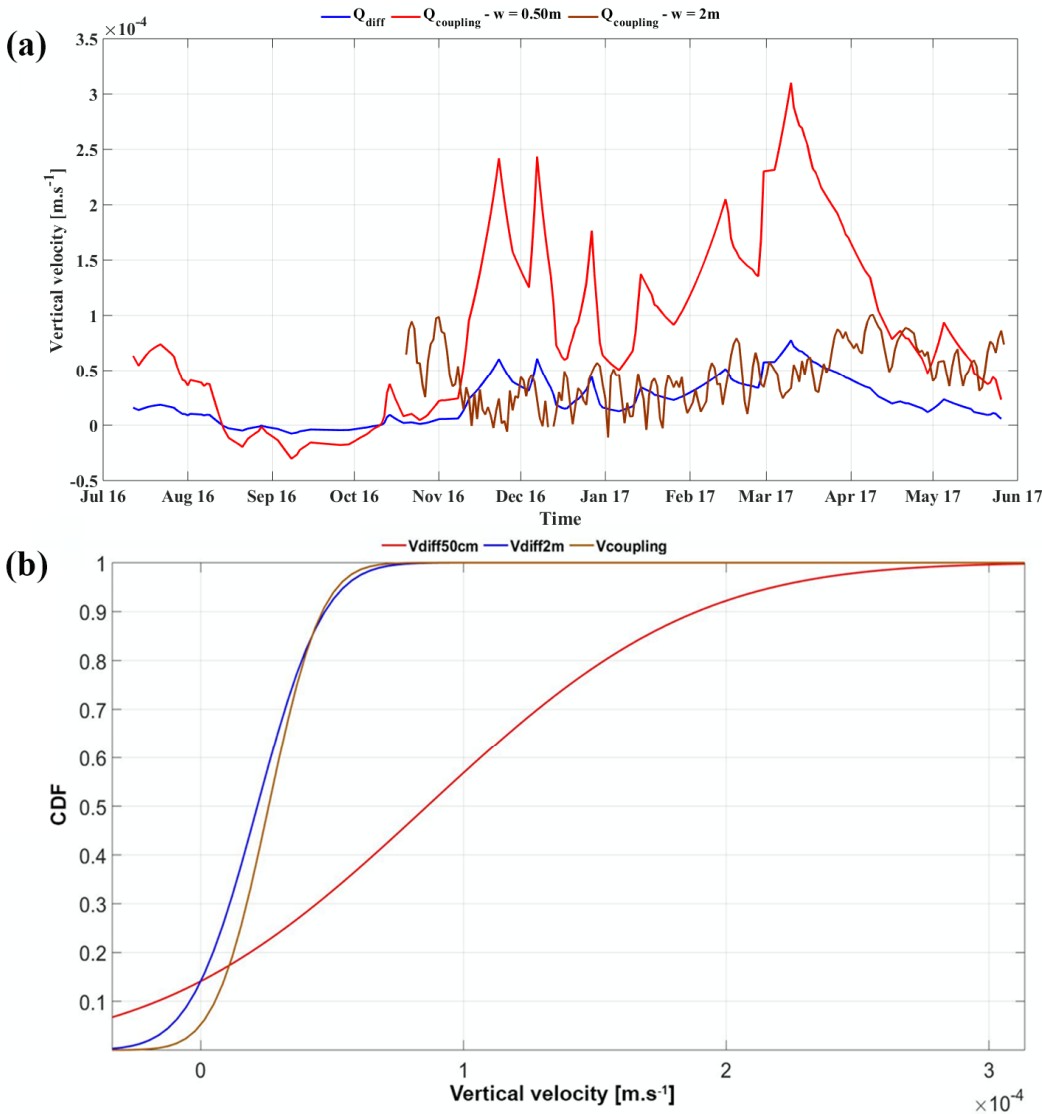

**Figure 10.** Linear velocities observed from differential gauging (V<sub>diff</sub>) normalized using a stream 2 m or 0.5 m wide and mean vertical flow (V<sub>coupling</sub>) calculated by coupling fiber optic distributed temperature sensing and thermal lances. (**a**) Dynamics of hyporheic flow over time. (**b**) Cumulative distribution function (CDF) of each hyporheic flow.

### 3.6. Limits of Our Method and Perspectives

In this study, we attempted to quantify exchanges between groundwater and surface water in a 400 m long reach by calculating vertical flow velocities in the sediment at every point of a FO-DTS cable. Given the large amount of data available, we relied on a few representative vertical flow velocities that we applied to the entire reach. Locations mapped as influenced (GW inflow) or not (neutral) by groundwater inflows (Figure 3b) were thus used to assess the spatial and temporal heterogeneity of groundwater–stream water exchange along the reach. Since only one fiber optic cable was buried in the sediment, but two temperatures at different depths are necessary to infer vertical flow, we coupled FO-DTS data with punctual data from thermal lances.

Generally, our coupling method provided mixed results that depended greatly on the area over which the aquifer and stream were assumed to exchange water. Since longitudinal heterogeneity was

restricted to the FO-DTS sampling distance (0.25 m), width became the adjustable variable of this exchange area. Thus, during the low-flow period, our method provided acceptable results as long as the stream was assumed to be narrow (0.50 m). Assuming a wider stream (2 m) yielded results that disagreed with low-flow observations and led to overestimation of hyporheic discharge. In contrast, at the end of the high-flow period (May 2017), volumetric estimates were more accurate when we assumed a wide area (2 m), since the stream occupied the entire streambed during high flow. Nevertheless, during most of the high-flow period, our method underestimated net hyporheic discharge greatly. Although we cannot rule out the possibility that vertical flow velocities were underestimated or did not represent the actual groundwater discharge, some clues indicate other factors at play. First, flow calculated from differential gauging discharge was generally similar to that of our method during the high-flow period. In addition, vertical velocities calculated above groundwater inflows were relatively homogeneous across the site. Finally, the locations where the velocities were calculated were chosen for their constancy over time and their clearness on the map, which indicated probable strong inflows that were confirmed by velocity results. These results make it unlikely that we underestimated groundwater flow velocity.

Instead, we attribute the large underestimation of discharge during high flow mainly to the small spatial coverage of our FO-DTS system. Our only cable was located in the thalweg and buried in the sediment to detect even the weakest inflows. This strategy paid off during low flow: As long as the stream was narrow, the groundwater inflows detected by the cable were likely to be the main ones. By assuming an area with a width equal to the actual stream width, we estimated total discharge similar to that from differential gauging. During the high-flow period, however, the stream broadened, and new flow paths appeared in the sediment that we did not detect. To attempt to account for undetected inflows at the bottom of the streambed, we widened the area (2 m) of the previously detected inflows. Unfortunately, this approach assumed that the groundwater–stream exchange is uniform over the 2 m, which is false: Some points considered neutral could have been near a strong inflow that went undetected, and vice versa [71]. In addition, studies have shown that many inflows are located directly in stream banks [72–75]. Our experimental design could not identify such lateral inflows. Furthermore, field observations showed that the banks were often flooded by groundwater surges that occasionally flowed directly into the stream. In late spring, these many lateral inflows and bank overflows probably weakened due to evapotranspiration [69], and inflows at the bottom of the streambed dominated again, probably explaining why our method yielded better estimates during this period (Figure 9).

Ultimately, our method was able to estimate total hyporheic discharge of a small reach as long as the areas of groundwater inflows and neutral points are estimated accurately. In this context, installing additional cables in the streambed and along the banks appears necessary. In parallel, calculating vertical velocities for only a few locations and then applying them to all similar points along the reach yielded relatively good estimates when the spatial coverage was sufficient. However, this method also has limits. First, it requires previously mapping groundwater inflows (see Part I of this study [40]). Second, calculating total hyporheic discharge from so few velocities makes it sensitive to hydrological processes affecting them, such as punctual inversions of the hydraulic gradient [76]. One solution would be to combine an increase in spatial coverage with automatic estimation of vertical velocity at each location of the cable, which might improve estimates of total hyporheic discharge greatly. This solution could also help to distinguish inflows that are sensitive to the local hydraulic gradient from less sensitive inflows that follow deeper regional flow paths [77,78].

## 4. Conclusions

Coupling FO-DTS and temperature depth profiles to study groundwater–surface water interactions is quite challenging. This research outlines a framework to quantify groundwater inflows too weak or diffuse to be characterized by traditional FO-DTS methods. We coupled FO-DTS data in the sediment with punctual temperature data at the water–sediment interface to estimate vertical flow velocities along the entire cable. Velocities of representative points were calculated per sub-reach and applied to similar points along the cable. Comparison of these high spatial resolution

velocities to those of spatially integrative methods revealed similar estimates of hyporheic discharge during the low-flow period but large underestimates during most of the high-flow period. This was attributed mainly to the limited coverage of the fiber optic cable, which could not detect lateral inflows during the high-flow period. Our method was also sensitive to punctual inversions of the hydraulic gradient that lead to slow vertical velocities.

Calculating vertical flows along the cable will improve the accuracy of the method and consequently discharge estimates during high flow. The high spatio-temporal resolution could help to distinguish groundwater inflows sensitive to local hydraulic changes from inflows that follow deeper regional flow paths or have different behaviors. Fine characterization of hyporheic exchanges (e.g., dynamics of deep inflows vs. lateral inflows, detection of downward flows) might be greatly useful for managing streams and studying hydro-biochemical processes.

**Author Contributions:** Conceptualization, H.L.L. and Z.T.; Methodology, H.L.L., F.R., P.P. and Z.T.; Software, H.L.L.; Validation, H.L.L., Z.T. and F.M.; Formal Analysis, H.L.L. and Z.T.; Investigation, H.L.L., F.R., P.P. and Z.T.; Resources, H.L.L., F.R., P.P. and Z.T.; Data Curation, H.L.L. and Z.T.; Writing-Original Draft Preparation, H.L.L. and Z.T.; Writing-Review & Editing, Z.T.; Visualization, H.L.L.; Supervision, Z.T. and F.M.; Project Administration, Z.T. and F.M.; Funding Acquisition, Z.T. and F.M.

**Funding**: This research was funded by Agence de l'Eau Loire Bretagne, grant number [150417801].

**Acknowledgments**: Authors warmly thank Mr. and Mrs. Pitois for the shelter and electricity they kindly provided during the entire campaign. We also thank all technicians from UMR INRA AGROCAMPUS OUEST SAS who helped us place the fiber optic cable and perform the field work. We also thank the ILSTER Zone Atelier Armorique.

**Conflicts of Interest:** The authors declare no conflict of interest.

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
