# Peer review of "Characterization of Diffuse Groundwater Inflows into Stream Water (Part II: Quantifying Groundwater Inflows by Coupling FO-DTS and Vertical Flow Velocities)"

_water, doi:10.3390/w11122430_

Round 1
Reviewer 1 Report
In this manuscript, the authors explored the quantification of groundwater inflows by coupling FO-DTS and vertical flow for the characterization of diffuse groundwater inflows into stream water. Overall, the manuscript is well written. The storyline is clear. The methods are solid. The analyses and discussion are decent.
Yet some following concerns similar to part 1 manuscript shall be addressed before potential publication.
Introduction: the authors shall elaborate the knowledge gap(s) along with the novelty and originality of this study in a more straightforward way.
Methods: the authors shall further justify the reason why this certain time period has been chosen. Why here not full year taken into account? Also, in my opinion, a roadmap of the framework shall be presented.
Discussion: I would like to suggest that the authors move the content regarding the limitations and the way forward/applicability in the present Conclusions section to the end of the Discussion section, and give more substantial information.
Author Response
Dear reviewer,
Thank you for reviewing our manuscript. We greatly appreciated your comments and suggestions. We have considered all the comments and we have worked thoroughly on a new version of the manuscript to tackle them all. As suggested by the reviewers, the abstract, introduction and materials were deeply revised. We also improved the figures, especially Fig. 1-6-7-8-9-10). Bibliography was completed, we added 4 papers from Water ([45, 46, 63, 65]).
Please find below our point-by-point responses to the reviewers’ comments. Our answers are in bold and modified sentences or passages are added. We refer to the specific line number of the modified manuscript, which is attached.
Thank you for your time.
Zahra Thomas on behalf of co-authors
Reviewer 1
Open Review
Comments and Suggestions for Authors
In this manuscript, the authors explored the quantification of groundwater inflows by coupling FO-DTS and vertical flow for the characterization of diffuse groundwater inflows into stream water. Overall, the manuscript is well written. The storyline is clear. The methods are solid. The analyses and discussion are decent.
Thank you for your kind feedback.Yet some following concerns similar to part 1 manuscript shall be addressed before potential publication.
Introduction: the authors shall elaborate the knowledge gap(s) along with the novelty and originality of this study in a more straightforward way.
Thank you to point it out, we have clarified it in the text. The introduction was rewritten:Lines 40-65: “ In their review of hydrologic exchange, Harvey and Goosef [9] outlined challenges in characterizing hydrologic connectivity, exchange flows, and related hydroecological processes. Tools from hydrological and ecological communities need to be combined to understand groundwater effects on stream flow and predict its future change. Many techniques such as seepage meters, mini-piezometric analysis [10, 11], differential gauging [12, 13], and chemical tracing [14, 15] have been used to quantify groundwater inflow into streams. Each of these techniques has a specific spatial range, accuracy, and limitations [16-20]. Most have the disadvantage of being punctual, such as seepage meters giving information only for a given radius (<1 m). Moving from a fine scale to a coarser scale requires spatially integrative methods such as tracing in specific wells, which sample a large volume and obtain average concentrations, but do not allow for detailed characterization of spatial heterogeneities. Differential gauging allows for measurements at the reach level (a few m to km) but does not allow for detailed description along the longitudinal profile. Also, it requires additional measurements when considering effects of tributaries. Among these techniques, temperature has been used to trace interactions between groundwater and surface water [21-23]. Since groundwater discharge affects the heat budget of the stream, its temperature can be modeled by solving a 1D transient advection-dispersion equation (Eq. 1) that requires stream temperature ( ), flow velocity ( ), dispersion coefficient ( ), net heat flux ( ), specific heat of water ( , the density of water ( , and the average water column ( . Variable t is the time and z is the distance along the direction of flow.
(1)
The net heat flux term includes the processes related to stream discharge, atmosphere, groundwater inflow and streambed conduction. Unidimensional analytical solutions of Eq. (1) have been developed and used for decades to infer groundwater inflow from thermal profiles in the sediment [24].”
Lines 84-86: “To address the challenge of quantifying weak inflows at a high spatial resolution, Mamer and Lowry [42] used FO-DTS to calculate vertical flow velocities of groundwater inflows through the hyporheic zone.”
Lines 94-101: “Because of the spatial variability of groundwater inflows, information obtained from punctual (a few cm) or integrative (a few km) methods are difficult to interpret. Multi-scale approaches combining multiple measuring methods may considerably constrain estimates of fluxes between groundwater and surface water [44-46]. In part I of this two-part study [40], we developed a framework to locate and map groundwater inflow along a reach. Here, we focus on quantifying groundwater inflows, which is essential for investigating resilience of aquatic ecosystems to climate change [47-49]. In this research, vertical flows in the hyporheic zone along the stream were calculated to infer groundwater discharge into the stream »
Lines 109-110: ” Our research framework consists of two parts to map and quantify diffuse and intermittent groundwater inflows into the stream.’
Methods: the authors shall further justify the reason why this certain time period has been chosen. Why here not full year taken into account? Also, in my opinion, a roadmap of the framework shall be presented.
The IUT lances were installed in October, that’s why we took the period from October to June for coupling FO-DTS and thermal lances. See lines 142-153: “Thermal profiles in the hyporheic zone [54, 55] were obtained using two types of temperature lances. The first were traditional hand-made lances made from a double PVC tube and fine mesh with TidbiT v2 sensors at four depths: +10, -30, -65 and -100 cm. With an accuracy of ±0.2°C and a time-step of 5 min, they gave broad estimates of interactions between stream and groundwater in the wetland and immediately before the confluence (Fig. 1c). One of the sensors (-30 cm) in the wetland broke down during the campaign, resulting in an incomplete profile. Because of the scarcity of our vertical data and their low spatial resolution and accuracy, three thermal lances of the second type (Umwelt-und Ingenieurtechnik (UIT) GmbH, Germany) were installed. These lances provided thermal profiles with high spatial resolution (+5, 0, -10, -20, - 35, -55, -85 and -115 cm) and good accuracy (±0.1°C) and resolution (0.04°C) (Fig. 1c). These new lances were installed in October 2016, ca. four months after installation of the FO-DTS system. Consequently, the period considered for this study extended from October 2016 to July 2017.”
Discussion: I would like to suggest that the authors move the content regarding the limitations and the way forward/applicability in the present Conclusions section to the end of the Discussion section, and give more substantial information.
Results and discussion were rewritten to focus on our findings. See highlighted text (too long to copy)
Reviewer 2 Report
I reviewed this MS as well as the authors' another MS (part1). I appreciate the authors' efforts to make better use of fiber optic cable data to estimate groundwater discharge velocity. The use of fiber optic cable undoubtedly drew much attention in recent years as it provides great spatial coverage of subsurface temperature measurements.
However, I didn't see the authors make full use of the fiber optic data as they simply averaged the measured data to separated segments, which make it no differences from traditional point measurements.
The authors directly use VFLUX2 to estimate the flux, which is also pretty common and lack of novelty in methodology.
Author Response
Dear reviewer,
Thank you for reviewing our manuscript. We greatly appreciated your comments and suggestions. We have considered all the comments and we have worked thoroughly on a new version of the manuscript to tackle them all. As suggested by the reviewers, the abstract, introduction and materials were deeply revised. We also improved the figures, especially Fig. 1-6-7-8-9-10). Bibliography was completed, we added 4 papers from Water ([45, 46, 63, 65]).
Please find below our point-by-point responses to the reviewers’ comments. Our answers are in bold and modified sentences or passages are added. We refer to the specific line number of the modified manuscript, which is attached.
Reviewer 2
Open Review
Comments and Suggestions for Authors
I reviewed this MS as well as the authors' another MS (part1). I appreciate the authors' efforts to make better use of fiber optic cable data to estimate groundwater discharge velocity. The use of fiber optic cable undoubtedly drew much attention in recent years as it provides great spatial coverage of subsurface temperature measurements.
However, I didn't see the authors make full use of the fiber optic data as they simply averaged the measured data to separated segments, which make it no differences from traditional point measurements.
FO-DTS maps of neutral and GW inflow locations were not averaged, to calculate the gradient. For each point we used the we used the temperature from FO-DTS of each 25cm dostance along the 400m length and we considered the temperature from the lances for each sub-reach as we do not have continuous measurements in the water column. The explanation was clarified to make this clear. Please refer to lines 230-240: “The UIT sensor set at the water-sediment interface (z = 0 cm) was used as the upper sensor for each calculation. At this depth, the sinusoidal diurnal signal has large amplitude. The FO-DTS point located above the representative groundwater inflow was used as the lower sensor for calculating flow (Fig. 2). At this depth, and because of the groundwater influence, the diurnal signal is clearly dampened (∆A) and greatly shifted in time (∆t) compared to that of the upper sensor (Eq. 3 and 4). The representative flow for neutral points was calculated directly using the UIT sensors at z = -10 cm. Its diurnal signal is not influenced by upward groundwater flow, so it is less dampened and phase-shifted. UIT lances were used for neutral points because the groundwater inflow mapping by FO-DTS (Part I) could not completely distinguish true neutral points in the sediment from uncovered cable. In contrast, UIT lances were placed in zones without clear inflow, and the depths of their sensors were known with certainty throughout the year. “The authors directly use VFLUX2 to estimate the flux, which is also pretty common and lack of novelty in methodology.
The novelty of the study lies in coupling FO-DTS and thermal profiles from vertical lances, to infer fluxes along the reach. Please refer to section 2.4.
Reviewer 3 Report
After inflow mapping in the first part, authors used vertical flow velocities in the second part of the companion manuscript. Authors coupled both of the methods because of some limitations to quantify the accurate groundwater inflows to streams. The second part of the manuscript determines the groundwater inflow by coupling the previous used method (FO-DTS) and vertical flow velocities for better accuracy and solid measurements. Authors described limitation of their method and restricted to only for low flows because the method underestimated the high flow hyporheic discharges. However, the results are clear and overall manuscript is well written. The manuscript follows the scope of the journal and could be interesting for readers. Following are some suggestion to the authors to improve some parts of the manuscript;
Section by Section:
Abstract:
Authors are suggested, beside discussing the methods please mention your key findings in the abstract section.
Introduction:
Please specify, how the second part is distinguished from the first one. It is suggested to highlight the impotence of the research, how the methods can be used to control stream water quality and base flows. Page 2-line 44-45; please specify the limitations and spatial and temporal ranges of the under study methods Page 2-line 48; please add some background research on “unidimensional analytical solutions used in the past Add main objectives of the research at the end of the section
Materials and Methods:
Page 3-line 106; Figure 1 is not so clear, please revise it. Try to add the grids defining latitudes and longitudes of the project place and scale bar as well. It is suggested to define VFLUX2 MATLAB toolbox and specify which input data was used to quantify the results? Give practical use of transport equation taking in to account your research. Suggest to consider both surface flow and seepage floe in soil (e.g. https://doi.org/10.5194/hess-23-4293-2019; https://doi.org/10.3390/ijerph16152729; https://doi.org/10.1007/s10040-019-01992-3; https://doi.org/10.1016/j.jhydrol.2019.123969; http://dx.doi.org/1061/(ASCE)GT.1943-5606).
Results and Discussion:
Page 7-line 272; it is suggested to mention the pattern of spatial and temporal temperature variation. Page 10-line 347; please illustrate, what does exactly mean the “Neutral locations”. On what criteria authors specified the neutral and inflow points? Page 13-line 434; please specify the three methods. Page 15-line 497-498; Authors stated that “Groundwater inflows and neutral/loss points were thus identified to assess the heterogeneity of the reach” is not so clear. Pease add more how the points were identified?
Conclusions:
Please summarize your innovative findings.
Scientific writing:
The scientific writing of the manuscript requires significant revision. I would like to suggest the manuscript to be professionally proofread and edited. Moreover, the authors may pay attention to some aspect of the conventional research writing, especially the connection between the sentences, the components/structure of the key parts (Abstract, Introduction, body, Conclusion).
Glasman-Deal, H. (2010). Science Research Writing for non-native speakers of English. Imperial College Press, London, 228p.
Author Response
Dear reviewer,
Thank you for reviewing our manuscript. We greatly appreciated your comments and suggestions. We have considered all the comments and we have worked thoroughly on a new version of the manuscript to tackle them all. As suggested by the reviewers, the abstract, introduction and materials were deeply revised. We also improved the figures, especially Fig. 1-6-7-8-9-10). Bibliography was completed, we added 4 papers from Water ([45, 46, 63, 65]).
Please find below our point-by-point responses to the reviewers’ comments. Our answers are in bold and modified sentences or passages are added. We refer to the specific line number of the modified manuscript, which is attached.
Thank you for your time.
Zahra Thomas on behalf of co-authors
Reviewer 3
Open Review
Comments and Suggestions for Authors
After inflow mapping in the first part, authors used vertical flow velocities in the second part of the companion manuscript. Authors coupled both of the methods because of some limitations to quantify the accurate groundwater inflows to streams. The second part of the manuscript determines the groundwater inflow by coupling the previous used method (FO-DTS) and vertical flow velocities for better accuracy and solid measurements. Authors described limitation of their method and restricted to only for low flows because the method underestimated the high flow hyporheic discharges. However, the results are clear and overall manuscript is well written. The manuscript follows the scope of the journal and could be interesting for readers. Following are some suggestion to the authors to improve some parts of the manuscript;
Thank you for your kind feedback.Section by Section:
Abstract:
Authors are suggested, beside discussing the methods please mention your key findings in the abstract section.
The abstract was rephrased and deeply revised to highlight our specific problem, our methodology and the main conclusions.Introduction:
Please specify, how the second part is distinguished from the first one. It is suggested to highlight the impotence of the research, how the methods can be used to control stream water quality and base flows.
The introduction was rewritten and completely rearranged and many statements that you pointed out before were addressed.Lines 40-65: “ In their review of hydrologic exchange, Harvey and Goosef [9] outlined challenges in characterizing hydrologic connectivity, exchange flows, and related hydroecological processes. Tools from hydrological and ecological communities need to be combined to understand groundwater effects on stream flow and predict its future change. Many techniques such as seepage meters, mini-piezometric analysis [10, 11], differential gauging [12, 13], and chemical tracing [14, 15] have been used to quantify groundwater inflow into streams. Each of these techniques has a specific spatial range, accuracy, and limitations [16-20]. Most have the disadvantage of being punctual, such as seepage meters giving information only for a given radius (<1 m). Moving from a fine scale to a coarser scale requires spatially integrative methods such as tracing in specific wells, which sample a large volume and obtain average concentrations, but do not allow for detailed characterization of spatial heterogeneities. Differential gauging allows for measurements at the reach level (a few m to km) but does not allow for detailed description along the longitudinal profile. Also, it requires additional measurements when considering effects of tributaries. Among these techniques, temperature has been used to trace interactions between groundwater and surface water [21-23]. Since groundwater discharge affects the heat budget of the stream, its temperature can be modeled by solving a 1D transient advection-dispersion equation (Eq. 1) that requires stream temperature (), flow velocity (), dispersion coefficient (), net heat flux (), specific heat of water (, the density of water (, and the average water column (. Variable t is the time and z is the distance along the direction of flow.
(1)
The net heat flux term includes the processes related to stream discharge, atmosphere, groundwater inflow and streambed conduction. Unidimensional analytical solutions of Eq. (1) have been developed and used for decades to infer groundwater inflow from thermal profiles in the sediment [24].”
Lines 84-86: “To address the challenge of quantifying weak inflows at a high spatial resolution, Mamer and Lowry [42] used FO-DTS to calculate vertical flow velocities of groundwater inflows through the hyporheic zone.”
Lines 94-101: “Because of the spatial variability of groundwater inflows, information obtained from punctual (a few cm) or integrative (a few km) methods are difficult to interpret. Multi-scale approaches combining multiple measuring methods may considerably constrain estimates of fluxes between groundwater and surface water [44-46]. In part I of this two-part study [40], we developed a framework to locate and map groundwater inflow along a reach. Here, we focus on quantifying groundwater inflows, which is essential for investigating resilience of aquatic ecosystems to climate change [47-49]. In this research, vertical flows in the hyporheic zone along the stream were calculated to infer groundwater discharge into the stream »
Lines 109-110: ” Our research framework consists of two parts to map and quantify diffuse and intermittent groundwater inflows into the stream.’
Page 2-line 44-45; please specify the limitations and spatial and temporal ranges of the under study methods
We have modified the text. See Lines 40-53: “In their review of hydrologic exchange, Harvey and Goosef [9] outlined challenges in characterizing hydrologic connectivity, exchange flows, and related hydroecological processes. Tools from hydrological and ecological communities need to be combined to understand groundwater effects on stream flow and predict its future change. Many techniques such as seepage meters, mini-piezometric analysis [10, 11], differential gauging [12, 13], and chemical tracing [14, 15] have been used to quantify groundwater inflow into streams. Each of these techniques has a specific spatial range, accuracy, and limitations [16-20]. Most have the disadvantage of being punctual, such as seepage meters giving information only for a given radius (<1 m). Moving from a fine scale to a coarser scale requires spatially integrative methods such as tracing in specific wells, which sample a large volume and obtain average concentrations, but do not allow for detailed characterization of spatial heterogeneities. Differential gauging allows for measurements at the reach level (a few m to km) but does not allow for detailed description along the longitudinal profile. Also, it requires additional measurements when considering effects of tributaries. »Page 2-line 48; please add some background research on “unidimensional analytical solutions used in the past Add main objectives of the research at the end of the section
We have added this new section to address your comment. We also added some new references to highlight the challenges on using multiscale approaches.Materials and Methods:
Page 3-line 106; Figure 1 is not so clear, please revise it. Try to add the grids defining latitudes and longitudes of the project place and scale bar as well.
Thank you for this comment. A new version of figure 1 presenting a 3D view of the study site with upstream hillslope cross section, longitudinal profile topography and stream cross section.It is suggested to define VFLUX2 MATLAB toolbox and specify which input data was used to quantify the results?
Input data for VFLUX2-Matlab were summarized Table 1. Sediment porosity, sediment thermal dispersivity, sediment thermal conductivity, sediment volumetric heat capacity and water volumetric heat capacity values were presented separately in Table 1 for each sub-reach.Give practical use of transport equation taking in to account your research. Suggest to consider both surface flow and seepage floe in soil (e.g. https://doi.org/10.5194/hess-23-4293-2019; https://doi.org/10.3390/ijerph16152729; https://doi.org/10.1007/s10040-019-01992-3; https://doi.org/10.1016/j.jhydrol.2019.123969; http://dx.doi.org/1061/(ASCE)GT.1943-5606).
We rearranged this part of the manuscript to give informations about heat equation. Please refer to lines 40-65:” In their review of hydrologic exchange, Harvey and Goosef [9] outlined challenges in characterizing hydrologic connectivity, exchange flows, and related hydroecological processes. Tools from hydrological and ecological communities need to be combined to understand groundwater effects on stream flow and predict its future change. Many techniques such as seepage meters, mini-piezometric analysis [10, 11], differential gauging [12, 13], and chemical tracing [14, 15] have been used to quantify groundwater inflow into streams. Each of these techniques has a specific spatial range, accuracy, and limitations [16-20]. Most have the disadvantage of being punctual, such as seepage meters giving information only for a given radius (<1 m). Moving from a fine scale to a coarser scale requires spatially integrative methods such as tracing in specific wells, which sample a large volume and obtain average concentrations, but do not allow for detailed characterization of spatial heterogeneities. Differential gauging allows for measurements at the reach level (a few m to km) but does not allow for detailed description along the longitudinal profile. Also, it requires additional measurements when considering effects of tributaries. Among these techniques, temperature has been used to trace interactions between groundwater and surface water [21-23]. Since groundwater discharge affects the heat budget of the stream, its temperature can be modeled by solving a 1D transient advection-dispersion equation (Eq. 1) that requires stream temperature (), flow velocity (), dispersion coefficient (), net heat flux (), specific heat of water (, the density of water (, and the average water column (. Variable t is the time and z is the distance along the direction of flow.
(1)
The net heat flux term includes the processes related to stream discharge, atmosphere, groundwater inflow and streambed conduction. Unidimensional analytical solutions of Eq. (1) have been developed and used for decades to infer groundwater inflow from thermal profiles in the sediment [24].
Results and Discussion:
Page 7-line 272; it is suggested to mention the pattern of spatial and temporal temperature variation.
We have modified the text to refer to “Normalized thermal anomalies mapped using the framework presented in Le Lay et al. (Part I in review) showed spatio-temporal variability observed over the study period (Fig. 3).”Page 10-line 347; please illustrate, what does exactly mean the “Neutral locations”. On what criteria authors specified the neutral and inflow points?
Locations without groundwater inflow are called “Neutral”, they are assumed to be influenced by the atmosphere. At neutral locations i.e. without inflow and assumed to be influenced by the atmosphere, vertical flow velocities. Please see lines 392-383:” At neutral locations (i.e. without inflow and assumed to be influenced only by the atmosphere), vertical flow velocities.”Page 13-line 434; please specify the three methods.
Thank you for this comment. The three methods (i) Qbalance from conservation of mass and energy (ii) Qdiff from differential gaging and Q coupling from coupling FO-DTS and vertical temperature profiles in hyporheic zone. We have modified the text lines 482-484Page 15-line 497-498; Authors stated that “Groundwater inflows and neutral/loss points were thus identified to assess the heterogeneity of the reach” is not so clear. Pease add more how the points were identified?
In this discussion we summarized our findings as presented lines 545-601“In this study, we attempted to quantify exchanges between groundwater and surface water in a 400 m long reach by calculating vertical flow velocities in the sediment at every point of a FO-DTS cable. Given the large amount of data available, we relied on a few representative vertical flow velocities that we applied to the entire reach. Locations mapped as influenced (GW inflow) or not (neutral) by groundwater inflows (Fig. 3b) were thus used to assess the spatial and temporal heterogeneity of groundwater-stream water exchange along the reach…..”Conclusions:
Please summarize your innovative findings.
The conclusions were clarified.
Scientific writing:
The scientific writing of the manuscript requires significant revision. I would like to suggest the manuscript to be professionally proofread and edited. Moreover, the authors may pay attention to some aspect of the conventional research writing, especially the connection between the sentences, the components/structure of the key parts (Abstract, Introduction, body, Conclusion).
Glasman-Deal, H. (2010). Science Research Writing for non-native speakers of English. Imperial College Press, London, 228p.
English was edited (see the certificate attached)

Round 2
Reviewer 2 Report
Accepted as it is.
Author Response
Dear reviewer,
Thank you for accepting our manuscript as it is.
Kind regards
Dr Zahra THOMAS
Reviewer 3 Report
There are some theoretical solutions for the groundwater flow into a closed barrier, please discuss the difference and similarity between the result from this study and the published solutions in literatures (Vilarrasa et al., 2011; Pujades et al., 2012, 2016, 2017; Shen et al., 2017; Wu et al. 2016; 2017; Wang et al., 2019; Xu et al., 2019). Please also discuss the effect of hydrogeology parameters.
Pujades E., et al (2012). Hydraulic characterization of diaphragm walls for cut and cover tunnelling. Engineering Geology. 125(27): 1-10.
Pujades E., et al (2016). Hydrogeological assessment of non-linear underground enclosures. Engineering Geology. 207 (3), 91–102.
Pujades, E., et al (2017). Settlements around pumping wells: analysis of influential factors and a simple calculation procedure. J. Hydrol. 548, 225–236.
Vilarrasa, V., et al (2011). A methodology for characterizing the hydraulic effectiveness of an annular low-permeability barrier. Engineering Geology, 2011, 120(1): 68-80.
Xu, Y.S., et al (2019). Experimental investigation on the blocking of groundwater seepage from a waterproof curtain during pumped dewatering in an excavation. Hydrogeology Journal, 27(7), 2659-2672. https://doi.org/10.1007/s10040-019-01992-3.
Wang, X.W., et al (2019). Evaluation of optimized depth of waterproof curtain to mitigate negative impacts during dewatering, Journal of Hydrology, 577(2019), 123969. https://doi.org/10.1016/j.jhydrol.2019.123969
Shen, S.L., et al (2017). Calculation of head difference at two sides of a cut-off barrier during excavation dewatering, Computers and Geotechnics, 91, 192-202. doi: 10.1016/j.compgeo.2017.07.014
Author Response
Dear reviewer,
Thank you for your kind recommandations. We have added a discussion about the limitation of the models especially its sensitivity to sediments properties and cite two main references on this topic (Irvine et al. 2015-WRR and Vandersteen et al. 2015-WRR).
The new text is in the paragraph where we presented sediment properties effects and limits lines 255-262 (the text of this presentation is going until line 272) : “Calculating vertical velocities based on analytical solution of 1D heat equation assumes that the fluid flow is vertical and one-dimensional. Temperature signal is assumed to be sinusoidal and there is no thermal gradient in sediment between the two sensors in the vertical direction [61]. As discussed in Irvine et al. [61], there is also a limitation regarding stream-bed heterogeneities which affect thermal properties of the sediments. Vandersteen et al. [62] developed a new method to calculate vertical flow called LPML. This method was compared to VFLUX. There were also major limitations with analytical solutions to the 1-D heat transport equation since streambed heterogeneity or non vertical flow components are not considered. In our study, we parameterized input…. »
Best regards
Dr Zahra THOMAS